# Characterization of the NiRAN domain from RNA-dependent RNA polymerase provides insights into a potential therapeutic target against SARS-CoV-2

Abhisek Dwivedy[1]☉, Richard Mariadasse[2]☉, Mohammed Ahmad[1]☉,
Sayan Chakraborty[1]☉, Deepsikha Kar[1]☉, Satish Tiwari[1], Sankar Bhattacharyya[3],
Sudipta Sonar[3], Shailendra Mani[3], Prafullakumar Tailor[1], Tanmay Majumdar[1]*,
Jeyaraman Jeyakanthan[2]*, Bichitra Kumar Biswal[1]*

**1** National Institute of Immunology, New Delhi, India, **2** Department of Bioinformatics, Alagappa University, Tamil Nadu, India, **3** Translational Health Science and Technology Institute, Faridabad, India

☉ These authors contributed equally to this work.
* majumdart@nii.ac.in (TM); jjkanthan@gmail.com (JJ); bbiswal@nii.ac.in (BKB)

**Data Availability Statement:** The raw data points for the graphs presented in Figs 5A, 6A, 6B and 6C; and the raw reads for the real-time PCR analysis

## Abstract

Apart from the canonical fingers, palm and thumb domains, the RNA dependent RNA polymerases (RdRp) from the viral order *Nidovirales* possess two additional domains. Of these, the function of the Nidovirus RdRp associated nucleotidyl transferase domain (NiRAN) remains unanswered. The elucidation of the 3D structure of RdRp from the severe acute respiratory syndrome coronavirus-2 (SARS-CoV-2), provided the first ever insights into the domain organisation and possible functional characteristics of the NiRAN domain. Using *in silico* tools, we predict that the NiRAN domain assumes a kinase or phosphotransferase like fold and binds nucleoside triphosphates at its proposed active site. Additionally, using molecular docking we have predicted the binding of three widely used kinase inhibitors and five well characterized anti-microbial compounds at the NiRAN domain active site along with their drug-likeliness. For the first time ever, using basic biochemical tools, this study shows the presence of a kinase like activity exhibited by the SARS-CoV-2 RdRp. Interestingly, a well-known kinase inhibitor- Sorafenib showed a significant inhibition and dampened viral load in SARS-CoV-2 infected cells. In line with the current global COVID-19 pandemic urgency and the emergence of newer strains with significantly higher infectivity, this study provides a new anti-SARS-CoV-2 drug target and potential lead compounds for drug repurposing against SARS-CoV-2.

## Author summary

The on-going coronavirus disease 2019 (COVID-19) pandemic caused by the severe acute respiratory syndrome coronavirus-2 (SARS-CoV-2) is significantly affecting the world health. Unfortunately, over 180 million cases of COVID-19 resulting in nearly 4 million deaths have been reported till June, 2021. In this study, using a combination of

are provided in the S1 Data file. The mass spectrometry raw data files are submitted to the OSF repository at- "https://osf.io/rt6aw/?view_only=815070ca377a4a6cb4aac2e56990152e". The results for the multiple sequence alignment, structural alignment, structural fold searches and small molecule dockings are submitted to the OSF repository at- "https://osf.io/bvcq2/?view_only=1a002bc15cf94bb884df06bc7d7601bc".

**Funding:** BKB was supported by the core funding from the National Institute of Immunology, Department of Biotechnology, New Delhi, India. JJ gratefully acknowledges Tamil Nadu State Council for Higher Education (File. No. RGP/2019-20/ALU/HECP-0049) for the financial support. The funders had no role in study design, data collection and analysis, decision to publish, or preparation of the manuscript.

**Competing interests:** The authors have declared that no competing interests exist.

bioinformatics, biochemical and mass spectrometry methods, we show that the Nidovirus RdRp associated Nucleotidyl transferase (NiRAN) domain of the RNA-dependent RNA polymerase (RdRp) of SARS-CoV-2 exhibits a kinase like activity. Additionally, we also show that few broad spectrum anti-cancer and anti-microbial drugs dampen this kinase like activity. Of note, Sorafenib, an FDA approved anti-cancer kinase inhibiting drug significantly reduces the SARS-CoV-2 load in cell lines. Our study suggests that NiRAN domain of the SARS-CoV-2 RdRp is indispensible for the successful viral life cycle and shows that abolishing this enzymatic function of RdRp by small molecule inhibitors may open novel avenues for COVID-19 therapeutics.

## Introduction

RNA dependent RNA polymerases (RdRp) are conserved across all RNA virus superfamilies harbouring a positive or negative sense RNA genome with the exception of retroviruses. RdRp catalyses the replication of RNA genomes from an RNA template, thus playing a key role in the viral life cycle and pathogenesis [1]. The structure of RdRp is broadly divided into three domains- fingers, palm and thumb, the shape resembling that of a right hand. This structural organization is conserved not only in RNA viruses, but also in the DNA polymerases across all kingdoms of life [1, 2]. However, the positive-stranded RNA (+RNA) viruses from the order *Nidovirales* present a conspicuous difference in the structural organization of their RdRp molecules [2–4]. The viruses in the order *Nidovirales* are known to infect a broad spectrum of hosts. While members of the families *Arteriviridae* and *Coronaviridae* infect vertebrates, members of *Mesoniviridae* and *Roniviridae* primarily infect invertebrates [2–4]. Despite a significant difference in the sizes of their genomes which ranges from 13 kb to 35 kb, the RdRp molecules encoded by each genome show a common structural organization [2].

Apart from the basic RdRp structural organisation comprised of the canonical palm, thumb and finger domains, the Nidovirus RdRp possesses two additional domains- the Nidovirus RdRp associated nucleotidyl transferase domain (NiRAN) and the interface domain [2–6]. The interface domain primarily serves as a connector between the NiRAN and canonical RdRp regions. However information on the functional aspects of the NiRAN domain remains limited. The first ever study that defines the NiRAN domain from the RdRp/Nsp9 of equine arteritis virus (EAV), demonstrates a manganese-dependent covalent binding of guanosine phosphate and uridine phosphate to an invariant lysine residue from the NiRAN domain [3]. The study proposes three possible molecular functions of the NiRAN domain: as a ligase, as a GTP dependent 5' nucleotidyl transferase and as a UTP dependent protein priming function facilitating the initiation of RNA replication [2, 3]. Various studies from other RNA viruses have demonstrated the nucleotidyl transferase activities exhibited by the N-terminal regions of the respective RdRp molecules [7–9]. However these transferase activities have been attributed to both 5'-priming as well as terminal ribonucleotide addition functions. Moreover, RdRp enzymes are known to exhibit either a primer dependent or a primer independent initiation of RNA replication [1, 10]. Interestingly, nidovirus RdRp molecules have been experimentally demonstrated to possess both modes of initiation, thus hinting at a probable UTP mediated priming function of the NiRAN domain [11, 12].

An earlier cryo-EM structure of replicase polyprotein complex from SARS-CoV (PDB ID-6NUR) had a significant portion of the NiRAN domain missing, thus failed to provide any functional association to this domain [5]. However, a recent study reporting the cryo-EM structure of the replicase polyprotein complex from SARS-CoV-2 (PDB ID- 7BTF, 6M71)

provides the complete structural preview of the NiRAN domain [6]. Together, these studies suggest that the interface domain also serves as the binding partner for Nsp8 protein of the polyprotein complex. COVID-19, the disease caused by the nidovirus SARS-CoV-2 is an on-going global crisis that necessitates the exploration of viral protein functions in addition to the exploration for novel anti-coronavirus inhibitor scaffolds [13–15]. In addition, the recent emergence of newer SARS-CoV-2 strains with significantly higher contagion [16] further necessitates the discovery of novel anti-viral targets and small molecule inhibitors targeting the life cycle and metabolism of the virus. In this study, utilizing a combination of *in silico* and bio-chemical tools, we propose that the NiRAN domain of SARS-CoV-2 possess a kinase like fold and is possibly involved in the catalysis of a kinase/phosphotransferase reaction. Moreover, we show that a number of previously known kinase and nucleotidyl transferase inhibitors dampen the activity of the NiRAN domain and of these; Sorafenib displayed anti-SARS-CoV-2 activity.

## Results

### Analysis of the NiRAN and Interface domains

As described in a recent study [6], the NiRAN and the interface domains span over residues 1–365 of the SARS-CoV-2 RdRp polypeptide sequence. These two domains form an arrow head like structure, which acts as a base for the RdRp region of protein (Fig 1A). Of these, amino acids 4–28 and 69–249 have been designated as the NiRAN domain that comprises of 7 α-helices (H1-H7) and 5-β strands (S1-S5), respectively. Two independent β-strands (B1, B2) form a hairpin like structure (residues 29–68) in the vicinity of NiRAN domain and are not considered integral to the NiRAN domain. The interface domain stretches from residue 250 to residue 365 and is comprised of a 6 α-helix bundle (h1-h6) followed by 3 antiparallel β strands (s1-s3). The overall topology suggests that the two distinct domains are connected by a small linker of 1 residue (Fig 1B). As proposed in the EAV- NiRAN study, several key residues such as K94, R124, S129, D132, D165 and F166 [3], were found conserved in the SARS-CoV-2-NiRAN as revealed in a pairwise alignment of the two sequences. The corresponding residues from SARS-CoV-2-NiRAN are K73, R116, T123, D126, D218 and F219. These residues have been predicted to be crucial for the enzymatic function of the NiRAN domain. A multiple sequence alignment of 17 RdRp polypeptide sequences from coronaviruses across multiple host species (human, bat, cow, pigs and rodents) revealed an absolute conservation of these residues (S1 Fig). Including the aforementioned residues, a total of 84 residues (45 in NiRAN and 39 in interface domains) were found strictly conserved in the RdRp molecules across all 17 viruses, hinting at a key role of these residues in the enzymatic and/or functional properties of NiRAN and Interface domains. A visualization of the conserved residues of the NiRAN domain shows that a significant majority of these residues are localized between the four stranded β-sheet and the following helix bundle (Fig 1B). A surface potential representation of the regions reveals the presence of five potential entry pockets suggesting that the active site of the NiRAN domain is possibly flanked by these conserved residues (S2A Fig). The 39 residues conserved in the interface domain are however spread all over the domain architecture (S2B Fig). The conservation of the residues across species suggests an important role of the NiRAN domain in RdRp mediated RNA replication in coronavirus, as suggested by the previous study on EAV-RdRp [3].

### Prediction of the plausible functions of the NiRAN domain

The understanding of the overall topology, localization of the conserved residues prompted an investigation on understanding the enzymatic function of the NiRAN domain. For this, two different approaches were considered with the Protein Data Bank as the target database. The

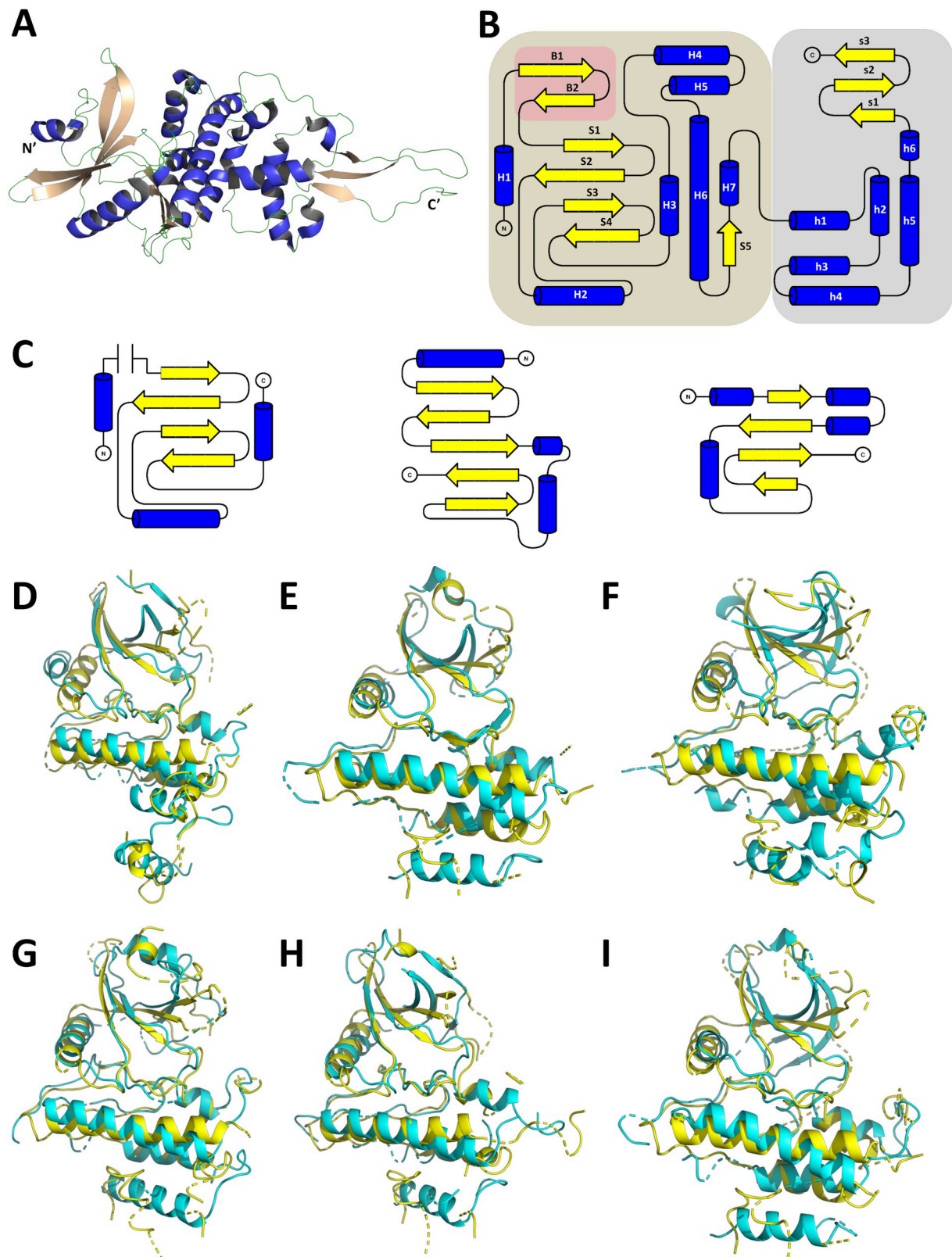

**Fig 1. The CoV-2-RdRp NiRAN domain presents a kinase/phosphotransferase like structural organisation.** (A) The NiRAN and the interface domains of the CoV-2-RdRp present an arrow head like structure (helices in royal blue, strands in peach and loops in green). (B) The

overall topology of the NiRAN (brown background, helices as marked as H and strands are marked as S) and interface domain (grey background, helices as marked as h and strands are marked as s) along with β-hairpin structure (brick red background, strands are marked as B). **(C)** The NiRAN domain possess a topology (left panel) that borrows elements from the canonical kinase fold (centre panel) as well as the non-canonical kinase fold of TgBPK1 (right panel). **(D-I)** The structural superimpositions of the secondary structural elements of CoV-2-RdRp NiRAN with known kinases reveal significant alignment in the antiparallel β-sheet and in the helices that follow. **(D)** Lim 2 kinase domain. **(E)** Syk kinase domain **(F)** O-mannosyl kinase domain. **(G)** IRAK4 kinase domain. **(H)** FGFR2 kinase domain. **(I)** Insulin receptor kinase domain (The aforementioned kinases' structural elements are shown in cyan, CoV-2-RdRp NiRAN structural elements shown in yellow).

first approach utilized the prediction of Hidden Markov Models (HHPred) [17] while the second approach predicted the presence of similar folds (ORION) [18]. The list of the top 15 hits retrieved for the first and second approaches are listed in S1 and S2 Tables, respectively. Notably, both approaches predicted a variety of kinase and kinase like transferase (phosphotransferase molecules). A previous report had evidenced at the similarity of the NiRAN domain to that of a known pseudokinase molecule SelO [5]; however further characterization was not feasible due to absence of the NiRAN 3D structure. We compared the topologies of the NiRAN domain with that of the canonical kinase fold. Similar to the canonical kinase fold, the NiRAN domain comprises an antiparallel β-sheet flanked by alpha helices. The canonical kinase domain exhibits a 5 stranded antiparallel β-sheet, which is flanked by two helices running parallel and one helix running perpendicular to the β-sheet (Fig 1C, middle panel) [19, 20]. However, the NiRAN domain shows a 4 stranded antiparallel β-sheet (S1-S4) flanked by one parallel (H2) and two perpendicular helices (H1 and H3) (Fig 1C, left panel). Further investigation into the available literature confirmed the presence of many non-canonical kinase folds, one of which presents a 4 stranded antiparallel β-sheet flanked by three parallel and one perpendicular helix (Fig 1C, right panel) [21]. Taken together, these results suggest that the NiRAN domain assumes a kinase like fold, possibly functioning either as a pseudokinase or a phosphotransferase.

To further investigate this proposition, a structural alignment of the NiRAN domain was performed against 8 randomly selected kinase molecules retrieved by the HHPred and ORION searches. The molecules Lim 2 kinase domain (PDB ID- 5NXD) (Figs 1D and S3A and S4A), Syk kinase domain (PDB ID- 4YJR) (Figs 1E and S3B and S4B), O-mannosyl kinase domain (PDB ID- 5GZ9) (Figs 1F and S3C and S4C) and IRAK4 kinase domain (PDB ID- 2NRU) (Figs 1G and S3D and S4D) aligned with the NiRAN domain with rmsd (root mean square deviation) values ranging from 0.6–4 Å and with alignment scores ≥ 0.5, suggesting that the kinase domains in these molecules share nearly similar fold with the NiRAN domain. As a proof of principle, two additional kinase molecules not listed in the aforementioned search results- FGFR2 kinase domain (PDB ID- 2PVF) (Figs 1H and S3E and S4E); and Insulin receptor kinase domain (PDB ID- 1GAG) (Figs 1I and S3F and S4F) were aligned with the NiRAN domain. Both these molecules displayed rmsd values ranging from 0.22–4 Å and alignment score of ~0.5, further highlighting the possible presence of a kinase like fold in the NiRAN domain. The results clearly exhibited the alignment of both the Cα backbone of the two molecules and the conservation of secondary structural elements. Of note, the alignments of the helix H2, helix H6, helix H8 and the antiparallel β-sheet (S1-S4) are strikingly evident. Of note, the previously mentioned pseudokinase molecule SelO (PDB ID- 6EAC) also aligned with the NiRAN domain. With rmsd values ranging from 2–8 Å and an alignment score of 0.42, the alignment suggests that though both the molecules roughly assume similar folds, the amino acid conservation differs significantly. Collectively, results of the alignments further highlight the presence of a non-canonical kinase/phosphotransferase like fold within the NiRAN domain.

## Predicting the NiRAN domain active site

The earlier study with EAV-RdRp experimentally demonstrated the binding of GTP and UTP nucleotides to the NiRAN domain [3]. A docking analysis of these nucleotides within the NiRAN domain was performed to delineate the probable active site. Kinase domains in general possess an active site between the antiparallel β- sheet and the subsequent helix bundle [19, 22, 23]. ATP (Fig 2A and 2A'), GTP (Fig 2B and 2B') and UTP (Fig 2C and 2C') docked well within the probable active site region with binding energies of -8.726 kcal/mol, -9.84 kcal/mol and -6.59 kcal/mol, respectively. Besides, end-point binding free energy calculation with MMGBSA (Molecular Mechanics Generalized Born Surface Area) suggested a strong interaction between the nucleotides and the residues in the probable active site (-17.84, -15.77 and -17.67 kcal/mol for ATP, GTP and UTP, respectively). Examination of the molecular interactions suggests that the ATP forms both salt bridges and H-bonding with key NiRAN residues such as K73 and D208. GTP formed possible H-bonding with R116 and salt bridge interaction with K73. The UTP however only displayed a salt bridge interaction with K73. Using the above interactions and the critically conserved residues, we computationally predicted the active site of NiRAN domain. The predicted active site pocket is lined with the following residues: F35, D36, I37, Y38, N39, F48, L49, K50, T51, N52, R55, V71, K73, R74, H75, N79, E83, R116, L119, T120, K121, Y122, T123, V204, T206, D208, N209, Y217, D218, G220, D221, F222 and S236; the underlined residues being strictly conserved across coronavirus RdRp molecules. K73 has been predicted as one of the key residues in the NiRAN active site. The corresponding lysine from the EAV-RdRp (K94) has been deemed critical to the function of NiRAN domain, owing to its interactions with GTP and UTP. The study also demonstrated that mutating lysine to alanine results in complete loss of RdRp function [3]. An overview of the active site pocket depicting the electrostatic surface potential indicates the localization of charged residues at the entry points to the site, while the interior of the pocket remains lined with largely non polar residues (S2C Fig).

With the above results suggesting a kinase like fold and the active site showing possible GTP and UTP binding function, a motif search was performed using the complete sequence of the NiRAN, β-hairpin and interface domains. The results predicted the presence of kinase like motifs belonging to Protein kinase C family, Protein kinase A family and Src Kinase family [24]. Few sequences motifs belonging to unspecified kinase families were also predicted. Interestingly, no kinase like motif was predicted for the interface domain and the β-hairpin region. Of note, a myristoylation consensus sequence overlapping with an unspecified kinase consensus was also predicted. Host enzymes are known to interact with viral proteins inducing post translational modifications following host or virus mediated signals. In particular, host kinases are known to post translationally modify viral protein often resulting in enhanced virulence or induction of various metabolic pathways [25–27]. A kinase site prediction suggested six different kinase phosphorylation sites with specificity for the eIF2 kinase family. A sequence based visualization of these motifs is presented in S5A Fig. These predictions hint at possible post translational modifications of viral proteins, mediated by the interplay of host–SARS-CoV-2, possibly aimed at regulating the host and viral metabolism.

In order to further explore the kinase like catalytic nature of NiRAN domain, three broad specificity kinase inhibitors- Sunitinib, Sorafenib and SU6656 were randomly selected and docked into the predicted active site of the NiRAN domain. All three kinase inhibitors show strong binding at the predicted active site as evident from the low free energy of binding (Fig 3A–3C and S3 Table). Interestingly, these inhibitors also demonstrate potential H-bond with residues lining the active site. While Sunitinib and Sorafenib primarily interact with aspartate residues, SU6656 exhibits H-bond interaction with lysine residue 73 (Fig 3A', 3B' and 3C'),

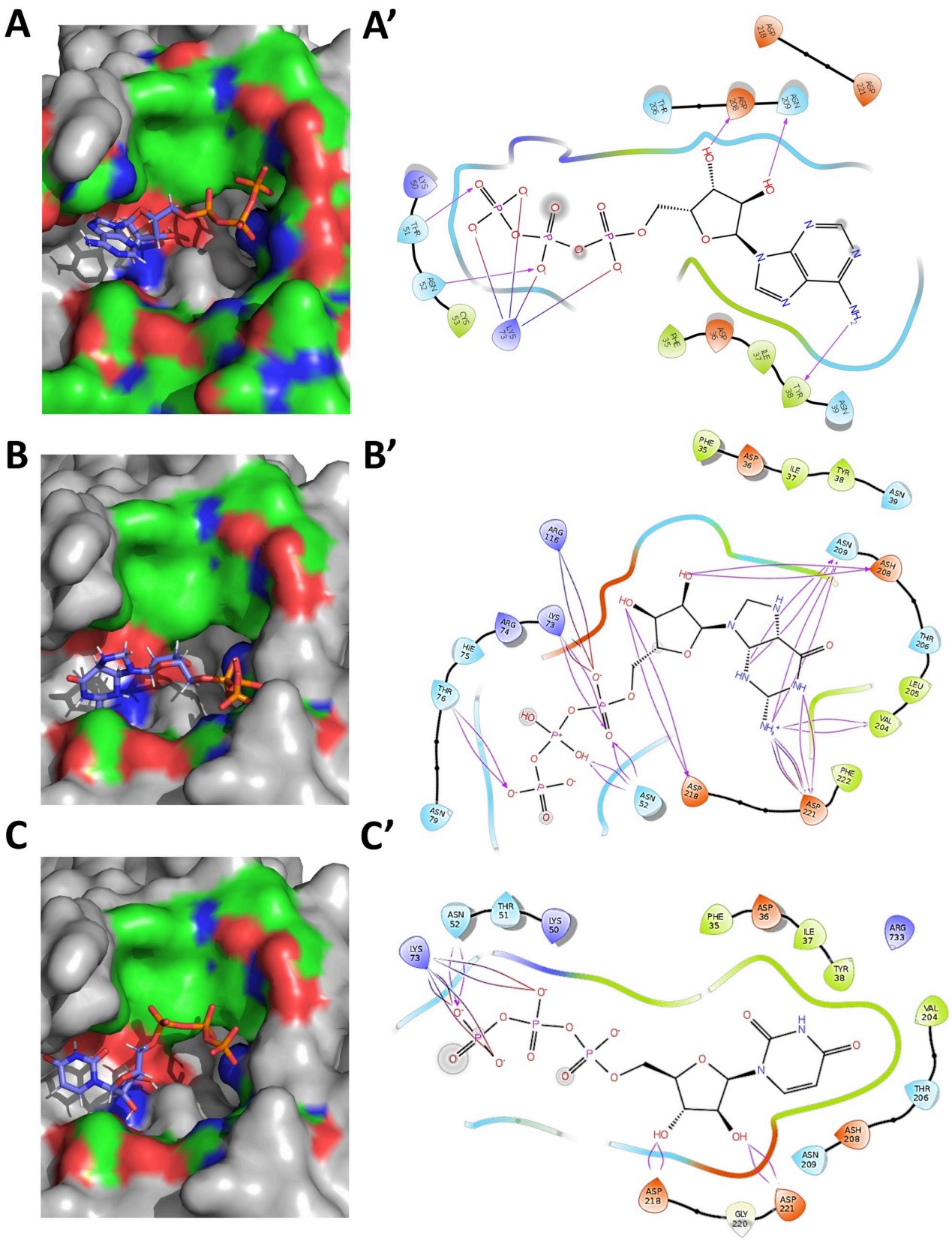

**Fig 2. The active site of CoV-2-RdRp NiRAN domain binds ATP, GTP and UTP and exhibits kinase like motifs.** ATP, GTP and UTP bind at the probable active site of the NiRAN domain with notably low free binding energies. **(A)** ATP within the active site pocket. **(B)** GTP within the active site pocket. **(C)** UTP within the active site pocket (Blue indicates positively charged regions, red indicates negatively charged regions and green indicates neutral regions, grey indicates regions beyond GTP-binding pocket). **(A')** ATP forms salt bridge with K73 and H-bonding with D208. **(B')** GTP binding reveals salt bridge interaction with K73 and H-bonding with Asp 116. **(C')** UTP binding exhibits salt bridge interaction with K73.

which is the critical lysine residue predicted to be involved in NiRAN domain's catalytic activity, as mentioned earlier. As a negative control the inactive analogs of two kinase inhibitors (Daidzein/PubChem ID- 5281708 and Geinstein/PubChem ID- 5280961) were docked into the putative NiRAN domain active site [28]. These compounds showed docking scores and binding energies lower than those of Sorafenib, Sunitinib and SU6656 (S5B and S5C Fig). In a nutshell, the above predictions further suggest that the NiRAN domain of coronavirus RdRp has a kinase/phosphotransferase like catalytic activity.

## Prediction of inhibitors targeting the NiRAN domain active site

As mentioned earlier, this domain might be involved in GTP induced protein phosphorylation, thus enabling a primer independent RNA replication (G-capping) [3]. Also, the domain may be involved in phosphorylation of UTP, thus functioning as a terminal nucleotidyl transferase. A wide range of viruses, both with DNA and RNA genomes are known to possess either multifunctional or dedicated proteins exhibiting the aforementioned activities [7, 29]. Interestingly, many pathogenic bacteria also encode for proteins that possess G-capping and/or terminal nucleotidyl transferase activities [30, 31]. Years of research have also presented a plethora of inhibitor molecules that target these essential functions in pathogens that cause diseases in human and non-human mammals [29]. After careful examination, a list of 77 compounds with experimentally demonstrated inhibitory potential against members of *Flaviviridae*, *Togaviridae*; Human cytomegalovirus, Herpes simplex virus, and few gram negative bacterial species were selected for docking against the CoV-2-RdRp NiRAN domain active site [29, 32–38]. The list of the selected compounds is presented in S4 Table.

Further, this study will discuss the best five inhibitors based on the docking analysis performed on the aforementioned 77 compounds. The five inhibitors with the best docking scores in a decreasing order are- 65482, 122108, 135659024, 4534, and 23673624 (numbers represent the PubChem IDs). The results of the docking analysis are presented in S5 Table. All these five inhibitors occupy varying regions within the active site pocket in a manner that their aromatic rings align with the uncharged/non-polar regions, while the charged moieties fit in the extremities of the active site pocket (Fig 4A–4E). All the five inhibitors exhibit H-bonding interactions with the active site residues. Interestingly all the five inhibitors present potential salt bridges with active site residues and also exhibit interactions with enzymatically critical residue K73, either via H-bonding or salt bridges. In addition to the above interactions, Compound 65482 also exhibits the pi-pi interactions between its own purine ring and residues F35 and F48, as is evident from its best docking score. The molecular interactions of these inhibitors with the CoV-2-RdRp NiRAN domain active site are presented in Fig 4A', 4B', 4C', 4D' and 4E'. In order to estimate the drug-likeliness of the aforementioned inhibitors, an ADME/T analysis was performed. The ADME/T analysis failed to provide any results for the compound 135659024. The ADME/T properties of the other four molecules were within the acceptable range in terms of Lipinski's rule of five [39] and related pharmacokinetic criteria such as H-bonding atoms probability, human oral absorption and the $IC_{50}$ values for blockage of human Ether-a-go-go-Related Gene (hERG) $K^+$. The results of this analysis are listed in S6 Table.

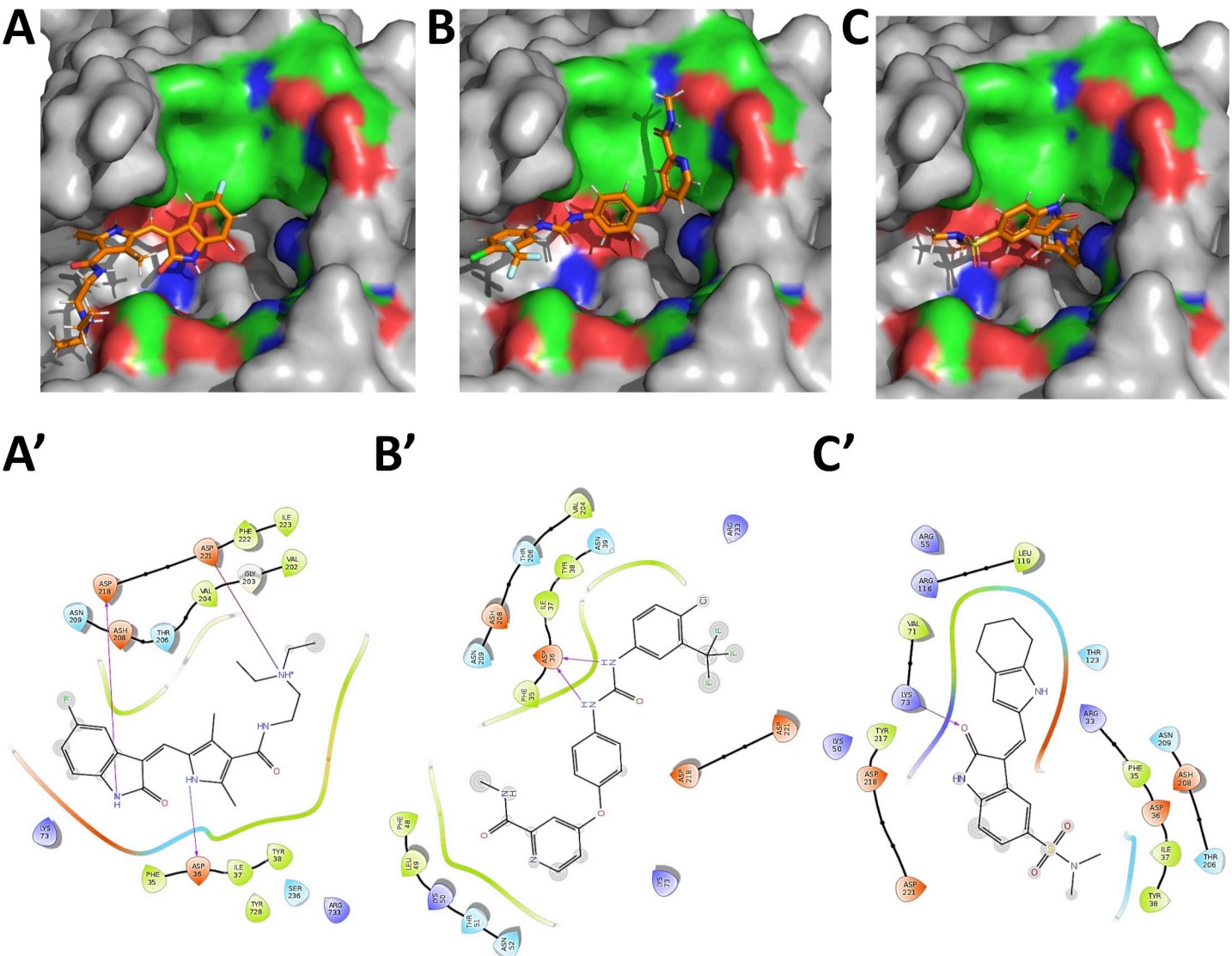

**Fig 3. The active site of CoV-2-RdRp NiRAN domain binds kinase inhibitors. (A, C and E)** Broad specificity kinase inhibitors bind within the predicted NiRAN active site with significantly low free binding energies. (Blue indicates positively charged regions, red indicates negatively charged regions and green indicates neutral regions, grey indicates regions beyond GTP-binding pocket): **(A)** Sunitinib within the active site pocket; **(B)** Sorafenib within the active site pocket; and **(C)** SU6656 within the active site pocket. The kinases inhibitors demonstrate H-bond interactions between with the enzymatically critical aspartate and lysine residues lining the active site- **(A')** Sunitinib; **(B')** Sorafenib; and **(C')** SU6656.

## The NiRAN domain of SARS-CoV-2 RdRp exhibits a kinase/phosphotransferase like activity

In order to determine any putative kinase like activity being harboured by the CoV-2 RdRp, the protein was overexpressed, purified (S6A and S6B Fig) and its identity was confirmed by mass spectrometry (S6C Fig and S7 Table). The absence of any known kinase/phosphotransferase substrate for CoV2-RdRp posed a different challenge. Most kinases are specific and rarely phosphorylate other targets [40]. In addition, in the absence of any substrate, kinases have been shown to exhibit a residual intrinsic ATPase/GTPase like activity [41–44]. Here, the enzyme enzymatically cleaves the nucleoside triphosphate molecule and transfers the gamma phosphoryl group to a water molecule in the microenvironment [44]. Also, majority of the

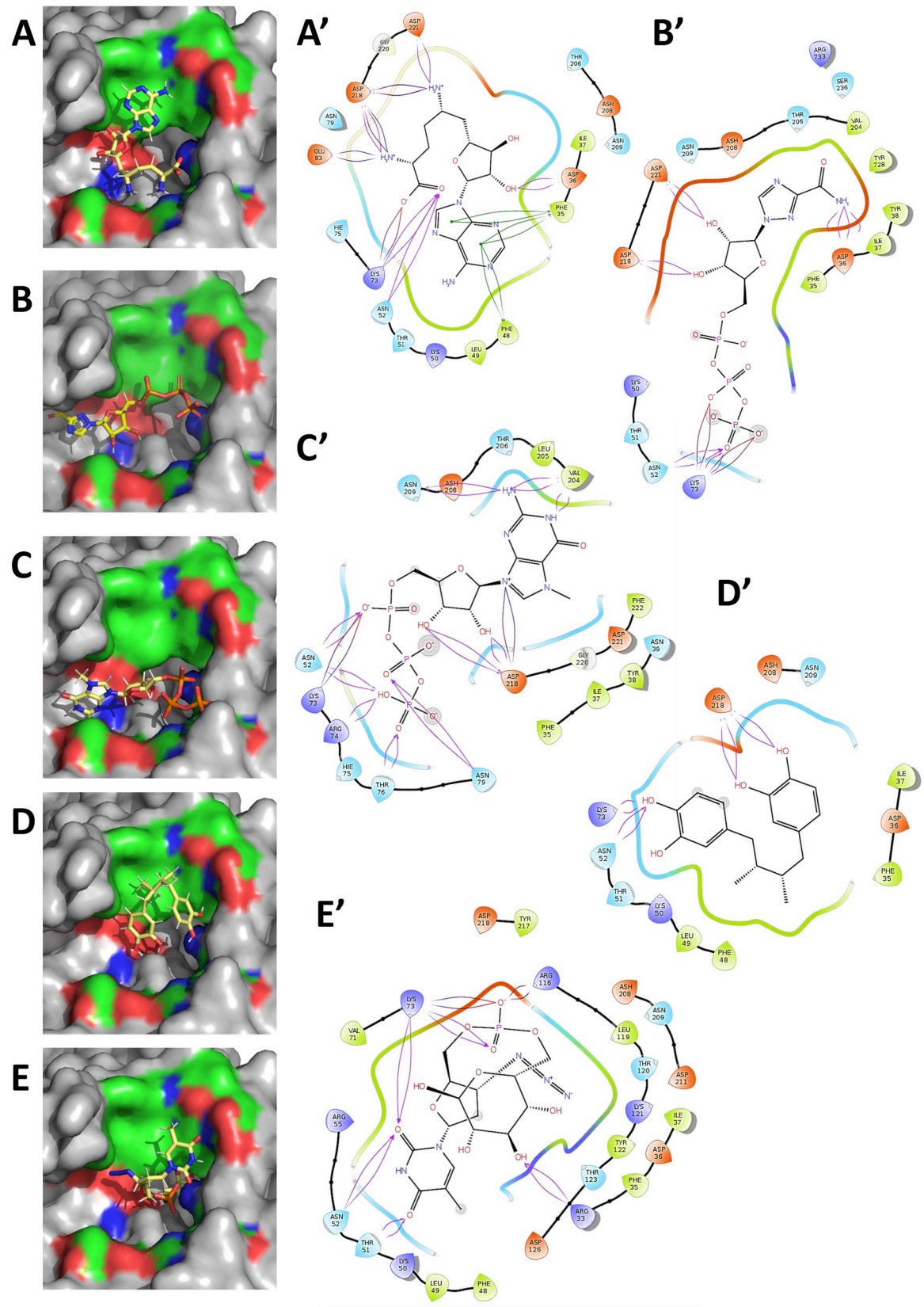

**Fig 4. The binding and molecular interactions of the best five predicted nucleotidyl transferase inhibitors at the active site of CoV-2-RdRp NiRAN.** The computationally directed binding of- **(A)** 65482; **(B)** 122108; **(C)** 135659024; **(D)** 4534; and **(E)** 23673624; within the active site pocket. (Blue indicates positively charged regions, red indicates negatively charged regions and green indicates neutral regions, grey indicates regions beyond GTP-binding pocket). The molecular interactions between the inhibitors and the active site pocket reveal H-bonds, salt bridges and pi-pi interactions- **(A')** 65482; **(B')** 122108; **(C')** 135659024; **(D')** 4534; **(E')** 23673624. Of note, the best predicted inhibitor- 65482 presents all the aforementioned molecular interactions.

known kinase inhibitors bind to the ATP binding site, thus abrogating the reaction [45]. An ADP-Glo Kinase assay kit [46] was utilized to determine the kinase activity of the CoV-2 RdRp. The efficiency of the assay in accurately determining a kinase activity was verified using a known kinase–human Akt2. The enzyme exhibited significant activity with a $K_m$ value of ~300 μM for ATP (Fig 5A), which is very similar to the $K_m$ values (~ 350 μM) determined in previously reported studies [47]. Also, Bovine Serum Albumin, a protein that binds ATP [48] exhibited negligible activity (Fig 5A), further verifying the specificity and selectivity of the protocol. Notably, incubation of the CoV-2 RdRp with varying concentrations of ATP exhibited significant activity akin to that of the human Akt2, with a $K_m$ of ~500 μM (Fig 5A). To further ascertain that the observed kinase/phosphotransferase like activity is primarily exhibited by the NiRAN domain, mutant proteins were generated where two key residues of the proposed NiRAN active site (K73 and D218) were mutated to alanines. A notable decline in the $V_{max}$ values of K73A and D218A mutants was observed (Fig 5A). However a double mutant for the key aspartate dyad of the RdRp replicase active site (D760 and D761 [49]) showed activity similar to that of the wild type enzyme (Fig 5A). Fluorescence spectrometry suggested that these mutations had no effects on the 3D structural integrity of the enzyme, suggesting that the loss of activity observed in K73A and D218A mutants are largely due to their role in the kinase/phosphotransferase type activity (S6D Fig). Together, these results present the first ever evidence of an intrinsic kinase/phosphotransferase like activity exhibited by the NiRAN domain of an RdRp molecule from coronaviruses.

To assess the efficiency of a phosphor-transfer reaction, dephosphorylated human histone H1 was treated with SARS-CoV-2 RdRp under appropriate conditions with appropriate controls. Mass spectrometry analysis of the trypsin digested histone H1 detected the presence of a phosphorylated peptide: 85- KS*LVS*KGTLVQTK-97 (phosphorylated at S86 and S89) following treatment with SARS-CoV-2 RdRp in presence of ATP and $Mg^{2+}$ (Figs 5B and S7A and S8A and S8 Table). Concurrently, the analysis of the histone H1 tryptic peptides detected the presence the same peptide phosphorylated at T92: 85- KSLVSKGT*LVQTK-97; following treatment with human Akt2 in the presence of ATP and $Mg^{2+}$ (Figs 5C and S7B and S8B and S8 Table). However, the same peptide from Histone H1 treated with RdRp or Akt2 in the absence of ATP; Histone H1 treated with ATP and $Mg^{2+}$ in the absence of RdRp or Akt2; and Histone H1 treated with only ATP and $Mg^{2+}$ had no phosphorylation in any of the serine and threonine residues (Figs 5D and S7C and S8C and S8 Table). These observations further suggest that the NiRAN domain of the SARS-CoV-2 RdRp has a serine-threonine like kinase activity.

## Inhibiting the NiRAN domain reduces viral load in ACE2 expressing cell lines

To further ascertain the possible kinase like activity of CoV-2 RdRp, both CoV-2 RdRp and human Akt2 were incubated with excess of ATP and treated with 500 nM of the each of the three kinase inhibitors mentioned in this study- Sorafenib, Sunitinib and SU6656. Interestingly, all the three kinases inhibitors significantly abrogated the kinase like activity of CoV-2 RdRp (Fig 6A). For human Akt2, Sorafenib and SU6656 significantly inhibited its kinase

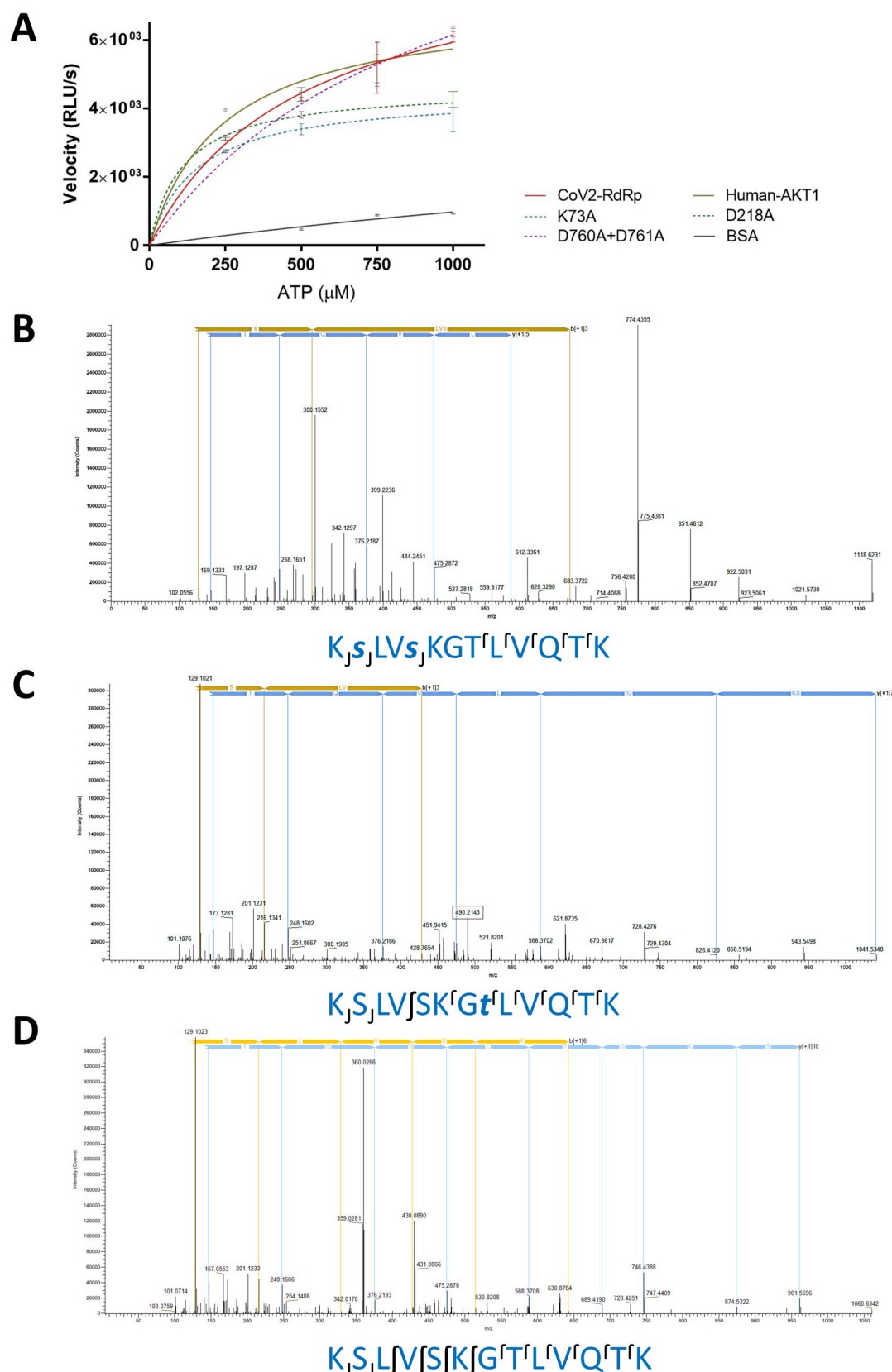

**Fig 5. SARS-CoV-2 RdRp exhibits a kinase/phosphotransferase like activity. (A)** The CoV-2 RdRp exhibits a kinase like activity akin to that of purified human Akt2. While mutations in the probable NiRAN active site (K73A, D218A) cause decline in the kinase like catalytic activity of the SARS-CoV-2 RdRp, mutating the RdRp replication active site (D760A+D761A) has negligible effect on the same. The ATP binding protein Bovine Serum albumin serves as a negative control (Data points show mean and standard error. The connecting curves represent non-linear regressions). **(B)** MS-MS spectra profile for peptide KSLVSKGTLVQTK from Histone H1 treated with SARS-CoV-2 RdRp in presence of ATP and $Mg^{2+}$ shows the phosphorylation at two serine residues. **(C)** MS-MS spectra profile for the same peptide from Histone H1 treated with human Akt2 in presence of ATP and $Mg^{2+}$ shows the phosphorylation at a single threonine residue. **(D)** MS-MS spectra profile for the aforementioned peptide from Histone H1 treated with SARS-CoV-2 RdRp in the presence of only $Mg^{2+}$ shows no phosphorylation, confirming the phosphotransferase activity of SARS-CoV-2 RdRp NiRAN domain (Further details are presented in S7 and S8 Figs and S8 Table).

activity, while Sunitinib treatment demonstrated a mild inhibition (Fig 6A). This provides a key insight into drug re-purposing for COVID-19, as all the three aforementioned compounds are well approved drugs for human usage with high tolerance and without any significant toxicity. As the NiRAN domain of CoV-2 RdRp has been annotated as nucleotidyl transferase, we sought to determine any inhibitory efficacy of the top five compounds predicted through *in silico* approaches. We were unable to procure the compound 23673624 commercially, thus we used the four remaining compounds- 135659024, 122108, 65482 and 4534. We observed very little effects of the first three compounds on the kinase like activity of CoV-2 RdRp at a concentration of 500 nM (Fig 6B). However, these compounds which are nucleoside analogs/derivatives exhibited conspicuous inhibitions at a concentration of 1000 nM (Fig 5C). The fourth compound 4534, a catechol derivative failed to inhibit the CoV-2 RdRp kinase like activity at any of the two concentrations (Fig 6B). This suggests that the three inhibitory compounds might be involved in blocking the binding of ATP to its respective binding site, in a manner similar to the specific kinase inhibitors.

In order to determine if inhibiting the activity of the NiRAN domain has any significant effect on the viral cycle of SARS-CoV-2, Vero E6 cells infected with SARS-CoV-2 were treated with the three broad spectrum kinase inhibitors at varying concentrations. Sunitinib and SU6656 failed to control the viral multiplication at the experimental concentrations. However, Sorafenib impeded the viral replication at concentration of above 2 μM, with efficiency similar to that of the known anti-viral drug, Remdesivir (at 1 μM) (Fig 6C). We observed an inhibition of viral growth at 5 μM of Sorafenib which is the minimum concentration of the drug that exhibited the maximum anti-viral efficacy. Sorafenib with 25 μM and 50 μM also encumbered viral multiplication, although there were no significant difference in viral load amongst the 5 μM, 25 μM and 50 μM concentration of the drug.

These results demonstrate that Sorafenib impedes the activity of NiRAN domain and thus hinders viral replication. Previous studies have shown that remdesivir abrogates the canonical replicase activity of RdRp [50, 51]. Notably, both palm and NiRAN domains amino acid sequences are highly conserved among various variants of concerns identified for SARS-CoV-2 [52, 53]. We hypothesized that a combinatorial therapy of remdesivir and sorafenib may show synergistic effect to abolish the dual enzymatic activities of RdRp. Furthermore, a combinational treatment of 5 μM Sorafenib and 1 μM Remdesivir shows a synergetic effect in neutralizing the virus (Fig 6C). Notably, Vero E6 cells exhibited no apoptosis when treated with Sorafenib at concentrations of 50, 25 and 5 μM and Remdesivir at concentrations of 25, 5 and 1 μM. These results suggest that Sorafenib alone or in combination with Remdesivir can prove effectively in clearing SARS-CoV-2 infections.

## Discussion

SARS-CoV-2, the causative agent of the on-going global pandemic COVID-19, is a recently emerged pathogen that has infected over 180 million and caused over 4 million deaths as of

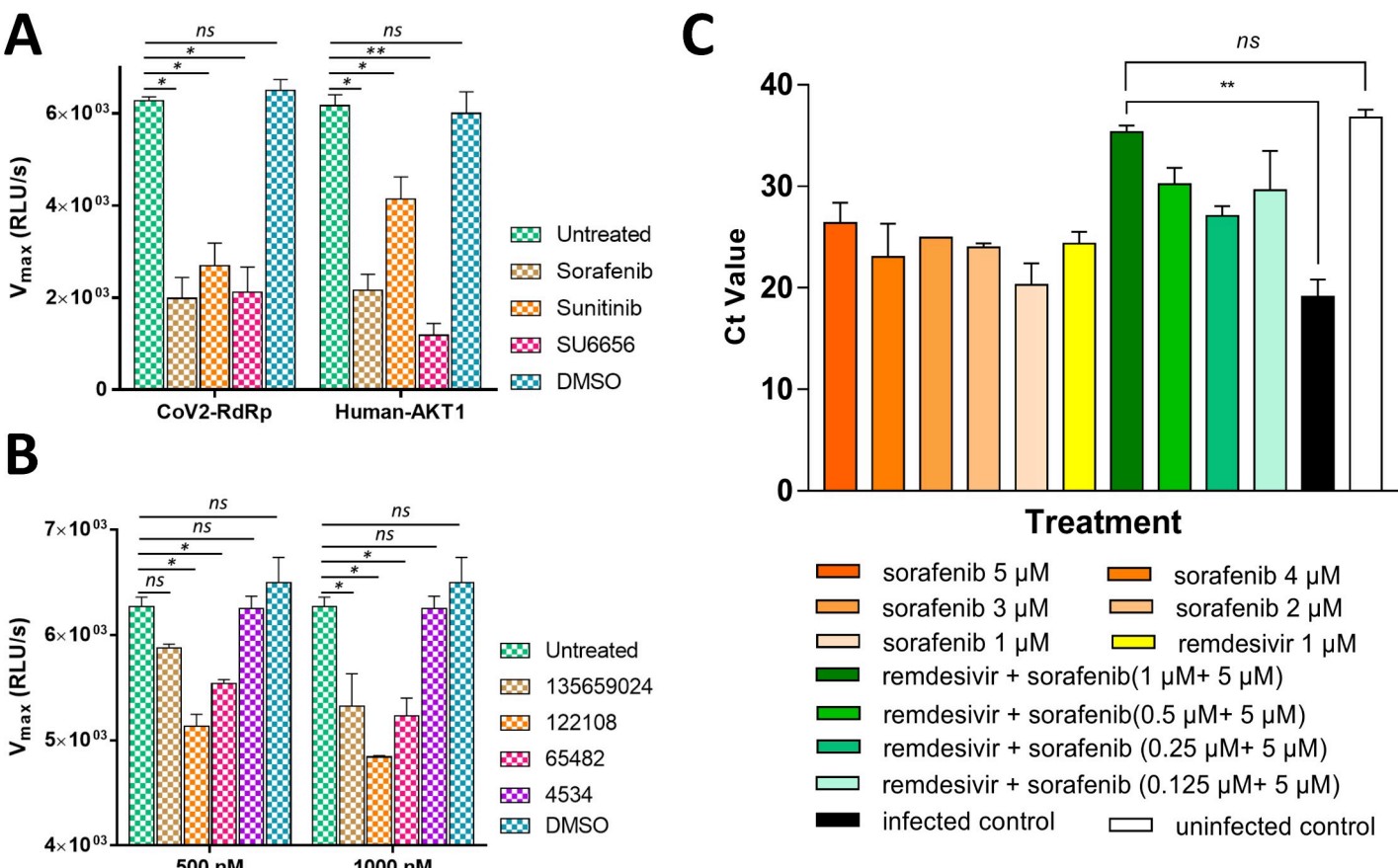

**Fig 6. Impeding the activity of the SARS-CoV-2 RdRp NiRAN domain reduces viral load in infected cells. (A)** While treatment with all the kinase inhibitors abrogate the kinase like activity of CoV-2 RdRp, the activity human Akt2 is majorly inhibited by Sorafenib and SU6656 with Sunitinib only exhibiting mild inhibition. **(B)** Treatment of CoV2-RdRp with nucleotidyl transferase inhibitors 135659024, 122108 and 65482 exhibit conspicuous inhibition of the kinase activity in micromolar concentrations. The compound 4534 however fails to exhibit any inhibitory potential. **(C)** 5 μM Sorafenib effectively reduces viral replication in SARS-CoV-2 infected Vero E6 cells. In addition, a combination of Sorafenib with 1 μM Remdesivir completely neutralizes SARS-CoV-2 as evident by a Ct value similar to that of uninfected cells (The bars represent mean and standard error. The symbols "*" and "**" represent significance for p-value less than 0.05 and 0.005, respectively. The symbol "*ns*" represents non-significance).

29th June, 2021 [54]. With a basic reproduction number ranging from 1.4 to 3.9, the disease has quickly disseminated across the globe within the past year [55]. The situation demands urgent attention from researchers worldwide in order to develop a better understanding of the viral pathogenesis, clinical presentations and biology of the disease in order to develop therapeutics such as vaccines and small molecule inhibitors targeting the viral proteins. Prevalent hypotheses suggest that multiple genome level recombination and zoonotic events between coronaviruses affecting human and bat host have resulted in the evolution of SARS-CoV-2 [15]. However it is worth noticing that the RdRp molecule of SARS-CoV-2 remains largely unchanged at the protein sequence level when compared to previous human coronaviruses such as SARS-CoV, MERS-CoV, SARS-Hku1 as well as non-human coronaviruses like Bat-CoV-Hku4 and Bat-CoV-ZC45, to name a few [56].

Despite being a well-known pathogen, following the SARS-CoV outbreak in 2002, the functional aspects of the RdRp molecule from coronaviruses remains unclear [5, 6, 11]. In addition, even less information is available on the peculiar NiRAN and interface domains that are specific to the RdRp molecules of viruses of the order *Nidovirales* [2, 3]. In this study, we propose that the NiRAN domain harbors an atypical kinase/phosphotransferase like fold, with

nucleoside triphosphate binding activity. The binding of the wide specificity kinase inhibitors at the predicted active site, further suggest the possible enzymatic action of the NiRAN domain. Interestingly, a recent report has suggested the possibility of kinase inhibitors in management of COVID-19 [57]. In line with the previous study on EAV-RdRp, this study delineates the possible catalytic residues at the NiRAN active site- F35, D36, F48, K73, R116, D218 and D221. In addition, the results also suggest that NiRAN domain of SARS-CoV-2 RdRp demonstrates a putative kinase/phosphotransferase like activity comparable to that of a well-established human kinase- Akt-1.

Using computational docking and simulation, this study predicts possible inhibitor compounds against the NiRAN domain. Interestingly, the aforementioned kinase like activity of the RdRp is significantly inhibited in the presence of these inhibitors in nanomolar concentrations. The compound Sunitinib is an inhibitor of the tyrosine family kinases [58]; Sorafenib inhibits activities of both serine/threonine and tyrosine family kinases [59, 60]; while SU6656 inhibits the Src family kinases [60, 61]. While Sorafenib and Sunitinib are approved for medical use in cases of renal, hepatocellular and gastro-intestinal cancers, the compound SU6656 is an experimental molecule used to study the role of Src kinases in cell cycle [61, 62]. In addition to the kinase inhibitors, the compound 65482 (Sinefungin), is broad specificity microbial nucleotidyl transferase inhibitor, and is known to inhibit RNA replication in flaviviruses and herpes viruses [63, 64]. The compound 122108 (Ribavirin 5'-triphosphate), inhibits the formation of g-capping of RNA in a wide range of viruses, such as Dengue virus, Hantaan virus and Hepatitis C virus [8, 65, 66]. Officially known as m7GTP, the compound with the PubChem Id 135659024, interferes with the RNA/DNA g-capping activity in many viral pathogens such as Rift valley fever virus, influenza virus, Zika virus and dengue virus, to name a few [67–69]. Given the unavailability of SARS-CoV-2 specific therapies and the emergence of newer strains [16], drug repurposing might prove crucial in combating the on-going epidemic while simultaneously being cost and time effective. This study examined the anti-SARS-CoV-2 activity of a well-studied drug, Sorafenib, which may prove effective in containing the diseases in moderate and severe cases of COVID-19. Other recent studies have hinted at the versatile nature of the NiRAN domain. Its catalytic functions include the NMPylation of nsp9 resulting in the formation of the replication transcription complex, essential for the viral replication [70]. Notably this NMPylation activity is specifically observed in the presence of $Mn^{2+}$, whereas the kinase/phosphotransferase activity presented in our study in observed only in the presence of $Mg^{2+}$ ions. In addition, another study has shown the role of NiRAN domain in formation of a cap core structure [71]. In addition, the NiRAN domains of the RdRp molecules from the various variants of SARS-CoV-2, including the variants of concern, exhibit high degree of conservation, especially in the putative active site residues such as K73 and D218 [52, 53]. Together our report and these recent studies hint at the indispensible nature of RdRp NiRAN domain for a successful viral replication in Nidoviruses, thus it is pertinent to explore *in vivo* anti-viral efficacies of compounds targeting the NiRAN catalytic functions.

## Materials and methods

### Ethics statement

All the experimental procedures were performed a minimum of three times with independently acquired proteins and cells lines. All experimental protocols were approved by the Institutional Bio-Safety Committees: IBSC#388/20 (approved for BKB by the National Institute of Immunology, India); IBSC#298/2021 (approved for SM by Translational Health Science and Technology Institute, India); and No.BT/Bs/17/458/2011-PID (approved for SM by Review Committee on Genetic Manipulation, Department of Biotechnology, India).

## Characterization of NiRAN and Interface domains of CoV-2-RdRp

The coordinates of the 3D structure of SARS-CoV-2-RdRp were retrieved from the Protein Data Bank (PDB IDs- 7BTF and 6M71) [6]. The dataset 7BTF was used for subsequent analysis as it provided information on all amino acid residues of CoV-2-RdRp. The coordinate data within the file was trimmed to amino acids 1–365 (sequence for NiRAN and Interface domains). This file was used for computational characterization of NiRAN and Interface domains. For sequence analysis, all the coronavirus RdRp sequences were retrieved from the NCBI database and subjected to multiple sequence alignment using the Clustal Omega [72] and Multalin [73] servers. The sequences alignment visualization was generated using the ESPript 3.0 web-tool [74]. Functional analyses were performed using the MPI Bioinformatics Toolkit [17], the DALI server [75] and the DSIMB server [18, 76]. The coordinate files for the kinases were retrieved from the Protein Data Bank (PDB IDs- 1CSN, 1GAG, 2NRU, 2PVF, 3Q60 and 6EAC) for structural alignment with the NiRAN and Interface domains of CoV-2-RdRp. The alignments were performed using the flexible "jFATCAT" algorithm in the Java application "strucalign" available at the Protein Data Bank [77]. The default alignments parameters were used (RMSD cut off- 3 Å, AFP distance cut off- 5, Maximum number of twists- 5, and Fragment length- 8). All visualization were performed in Pymol [78]. Motif prediction and kinase functional prediction were performed using the web servers- ELM [79], PSIPRED [80], CDD [81], MyHits [82], and PhosphoPredict [83].

## *In silico* ligand preparation, active site pocket prediction and molecular docking

The coordinate files for all the inhibitors were retrieved from the PubChem Database and/or from Protein Data Bank. All the inhibitors were prepared by assigning appropriate geometries, bond orders, tautomers and ionization states using LigPrep module implemented in the "Schrodinger Suite 2019" [84]. The coordinate file for the NiRAN and Interface domains was used to predict the probable ligand binding pockets using the SiteMap module of the "Schrodinger Suite 2019". The accuracy of the sites were verified using the Phyre [85] and CASTp servers [86] and the results were compared. These sites were used to generate a Grid for molecular docking of the inhibitors. The prepared inhibitors were docked onto the predicted binding sites using Glide XP (extra Precision) module of the "Schrodinger Suite 2019".

## MMGBSA Calculation

The free energies of binding for the inhibitor-domain complexes (NiRAN and Interface domains with inhibitors) were analysed by MMGBSA (Molecular Mechanics Generalized Born Surface Area) calculation [87]. The calculation was executed using the following equation:

$$G = \Delta E_{MM} + \Delta G_{SGB} + \Delta G_{SA}$$

($\Delta E_{MM}$ represents the minimized molecular energy changes in gas phase; $G_{SGB}$ represents the surface calculation using the GB model; and $G_{SA}$ represents the accessible surface area of the complexes) [88, 89].

The molecular docking and MMGBSA calculation protocols were validated by comparing with the available co-crystal structure of Zika Virus methyl-transferase bound to the nucleotidyl-transferase inhibitor Sinefungin (PDB ID- 5MRK).

## ADME/T properties prediction

The top 5 inhibitors with the highest docking scores and minimum binding energies were analysed for their pharmacokinetic properties such ADME (Absorption, Distribution, Metabolism, Elimination or Toxicity) and drug likeliness [90, 91]. The QikProp module of the "Schrodinger Suite 2019" was utilized to evaluate the drug likeliness by employing the Lipinski's rule of five [39] and various pharmacokinetic properties such as probable Hydrogen bonding atoms; Human Oral Absorption (<25%: poor; >above 80%: good); QP LogS (Predicted aqueous solubility of the compounds, acceptable range is -6.0 to -0.5) and the $IC_{50}$ values for blockage of the human Ether-a-go-go-Related Gene (hERG) $K^+$ channels (< -5: satisfactory) [92].

## SARS-CoV-2 RdRp native and mutant proteins generation

The ORF encoding for the SARS-CoV-2 RdRp was PCR amplified using gene specific primers (mentioned at the end of this section) and using the plasmid pDONR223 SARS-CoV-2 NSP12 as the template DNA (Addgene, USA). Codons encoding for a deca-histidine tag was incorporated into the forward primer in order to obtain the recombinant protein with an N-terminal deca-histidine tag. An entry cloning site (CACC), a recognition site for the restriction endonuclease *NdeI* (CATATG) and a start codon were also incorporated into the forward primer. The reverse primer included a stop codon and the recognition site for the restriction endonuclease *HindIII* (AAGCTT). The primers used are listed at the end of this paragraph. The PCR amplified product was ligated into the entry vector pENTR-D-TOPO using appropriate protocol (Qiagen, USA). The ligated product was transformed into chemically competent *Escherichia coli* strain DH5α and grown on LB agar medium containing 50 μg/ml kanamycin for selection at 310 K. Positive colonies were screened using colony PCR and recombinant plasmid pENTR-D-TOPO-RdRp was extracted by plasmid mini prep using appropriate protocol. The isolated plasmid was subjected to restriction digestion with the enzymes *NdeI* and *HindIII* using appropriate protocol (New England Biolabs, UK). The expression vector pET-28b (+) was linearized by restriction digestion with enzymes *NdeI* and *HindIII*. The digested fragment containing the ORF for RdRp and the linearized expression vector were ligated by T4 DNA ligase employing the appropriate protocol (Qiagen, USA). The ligated product was transformed into chemically competent *Escherichia coli* strain DH5α and grown on LB agar medium containing 50 μg/ml kanamycin for selection at 310 K. Positive colonies were screened using colony PCR and recombinant plasmid pET-28b-RdRp was extracted by plasmid mini prep using appropriate protocol. The plasmids were verified by restriction digestion with enzymes *NdeI* and *HindIII* and by DNA sequencing (BioServe, India). Single codon substitustions for K73A and D218A and double codon substitution for D760A+D761A were performed using a nested PCR based site directed mutagenesis protocol described earlier [93] using appropriate primers (mentioned at the end of this section). The mutant plasmids were confirmed by DNA sequencing (BioServe, India). The plasmids pET-28b-RdRp (native and mutants) were then into transformed into chemically competent *Escherichia coli* strain BL21-DE3 and grown on LB agar medium containing 50 μg/ml kanamycin for selection at 310 K.

A single colony was picked from the agar medium and inoculated into LB liquid medium containing 50 μg/ml kanamycin and grown at 310 K. Large scale liquid cultures were further inoculated from the aforementioned culture and were grown at 310 K. These cultures were induced with 0.5 mM of IPTG at an $A_{600}$ of 0.8. The cultures were maintained for 3 hours post induction at a temperature of 301 K. The cells were harvested by centrifugation of the cultures at 10000 g and were re-suspended in the lysis buffer (50 mM Tris, 500 mM NaCl, 30 mM

Imidazole, 2.5 mM DTT, 10% glycerol, PIC 1 tablet, 10 mM PMSF, 0.1% Nadeoxycholate, 0.3 mg/ml lysozyme at pH 8.0). The cells were lysed using a CF Cell Disruptor (Constant Systems, UK). The lysate was centrifuged at 12000 g for one hour and the supernatant was utilized for protein purification. The protein was purified using the fast performance liquid chromatography system AKTA Pure (GE Healthcare, USA). The protein was purified using an affinity chromatography (HisTrap-FF column; GE Healthcare, USA), followed by a size exclusion chromatography (HiLoad 16/600 Superdex 200 PG column, GE Healthcare, USA). In affinity chromatography the recombinant protein was eluted in the aforementioned buffer containing 300 mM imidazole (gradient 1 to 500 mM). In ion exchange chromatography the protein eluted in the aforementioned buffer containing an additional 210 mM NaCl (gradient 1 to 1000 mM). In size exclusion chromatography, the protein eluted at a flow volume of ~65 ml. The complete purification process was maintained at a temperature of 277 K. The purity of the recombinant protein was analysed on SDS-PAGE and the identity of the protein was confirmed using liquid chromatography- tandem mass spectrometry (Ultra-High Resolution Nano LC-MS/MS using C18 column and Q Exactive Orbitrap Mass Spectrometer, Thermo-Scientific, USA) at a commercial facility (VProteomics, New Delhi, India). The purified protein was concentrated to 1 mg/ml using a 50 kDa membrane centrifugal unit (Millipore, USA) and stored for further use. The native and the mutant RdRp proteins were over-expressed and purified using the same aforementioned protocol. Fluorescence spectra of the native and mutant RdRp proteins were acquired at 30˚C using a Fluorescence Spectrophotometer (Horiba Fluromax4, Japan) with an excitation wavelength of 280 nm, and the emission spectra of protein samples were recorded from 300 to 400 nm. All the spectra were corrected for buffer absorption and the proteins were diluted to a concentration of 5 µM.

| Primers | |
|---|---|
| Native RdRp | |
| Forward Primer | 5'-CACCCATATGAUGCACCATCACCATCACCATCACCACCCAA CTTTGTACAAAAAAGTT-3' |
| Reverse Primer | 5'-TATAAGCTTTTACTGCAGCACGGTGTGAGGGGTGTA-3' |
| RdRp K73A mutant | |
| Forward Primer | 5'- ATCCTGTCTGCCGCCGCGGTGGTG-3' |
| Reverse Primer | 5'- CACCACCGCGGCGGCAGACAGGAT -3' |
| RdRp D218A mutant | |
| Forward Primer | 5'- TTCGTGGTGGCGAGGCACACC-3' |
| Reverse Primer | 5'- GGTGTGCCTCGCCACCACGAA -3' |
| RdRp D760A+D761A double mutant | |
| Forward Primer | 5'- AACTGGTACGCCTTCGGAGA -3' |
| Reverse Primer | 5'- GTCTCCGAAGGCGTACCAGTT -3' |

## Biochemical Assays of SARS-CoV-2 RdRp native and mutant proteins

The ADP-Glo Kinase Assay system (Promega, USA) was utilized for all biochemical analysis using purified recombinant RdRp [46]. The enzyme concentration used for all biochemical assays was 100 nM. The buffer used for biochemical assay contained 20 mM Tris-Cl, 300 mM NaCl, 7.5 mM MgCl$_2$, 5 mM 2-mercaptoethanol and 10% v/v glycerol. Varying concentration of ATP (250, 500, 750 and 1000 µM) were utilized to determine the enzymatic activity of the recombinant RdRp at 310 K. The readouts were measured at 0, 5, 20, 35 and 50 minutes post

initiation of the reaction using a luminometer (Berthold Technologies, Germany). The same protocol was utilized for the biochemical assays of mutants K73A, D218A and D760A +D761A. Purified Human Akt2 (Sigma Aldrich, USA) at 100 nM and Bovine Serum Albumin at 100 μM (HiMedia, India) concentrations were used as positive and negative controls for Kinase activity, respectively.

The kinase inhibitors Sorafenib, Sunitinib and SU6656 (Sigma Aldrich, USA) were dissolved in 100% v/v DMSO (Sigma Aldrich, USA) and used at a concentration of 500 nM. The predicted nucleotidyl transferase inhibitors 135659024, 122108, 65482 and 4534 (Sigma Aldrich, USA) were dissolved in 100% v/v DMSO and used at two different concentrations- 500 and 1000 nM. For all inhibition studies, the ATP concentration was maintained at 100 mM. The graphs were plotted using GraphPad Prism 7.0. Non-linear regressions of the enzymatic activities were performed for obtaining the Michaelis-Menten kinetics. Statistical significance for the inhibitory potential of the compounds was determined using student's t-test.

## Mass Spectrometry analysis of the Kinase/Phosphotransferase activity of SARS-CoV-2 RdRp NiRAN domain

De-phosphorylated Histone H1 (Sigma Aldrich, USA) was treated with different combinations of SARS-CoV-2 RdRp or human Akt2, 1000 μM ATP and 7.5 mM $MgCl_2$; in a buffer containing 20 mM Tris-CL, 300 mM NaCl, 5 mM 2-mercaptoethanol and 10% v/v glycerol. The six reaction mixtures were set up: (1) Histone H1 + RdRp + ATP + $Mg^{2+}$; (2) Histone H1 + Akt2 + ATP + $Mg^{2+}$; (3) Histone H1 + BSA + ATP + $Mg^{2+}$; (4) Histone H1 + RdRp + $Mg^{2+}$; (5) Histone H1 + Akt2 + $Mg^{2+}$; and (6) Histone H1 + ATP + $Mg^{2+}$. The samples were incubated at 310 K for 1 h and the proteins were separated on SDS-PAGE. The subsequent protocols were carried out at commercial facility (VProteomics, New Delhi). Briefly, the bands of interest, corresponding to the size of Histone H1 were excised from the gel and were treated 4 mM of Tris (2-carboxyethyl)phosphine (Sigma Aldrich, USA) 8 mM iodoacetamide (Sigma Aldrich, USA). In-gel digestion was performed with uHPLC grade trypsin (Sigma Aldrich, USA) in 50 mM ammonium bicarbonate (Sigma Aldrich, USA) at 310 K for 12 h. Digests were cleaned using a C18 silica cartridge to remove the salt and dried using a speed vac. The dried pellet was resuspended in 5% acetonitrile (Sigma Aldrich, USA) and 0.1% formic acid (Sigma Aldrich, USA). Liquid chromatography tandem mass spectrometry was performed on an Ultimate 3000 RSLCnano system coupled with a Thermo Q-Exactive Plus. 1ug of sample was loaded on a C18- 50 cm, 3.0μm Easy-spray column (Thermo Fisher Scientific). Peptides were eluted with a 0–40% gradient of 80% acetonitrile, 0.1% formic acid; at a flow rate 5 nls$^{-1}$ and injected for MS analysis. LC gradient was run for 100 minutes. The mass spectrometer was driven in data-dependent acquisition mode using the Thermo Xcalibur 2.1 software for a full spectrum scan ranging from 150-1200m/z. MS1 spectra were acquired in the Orbitrap at 70k resolution. Dynamic exclusion was employed for 10 s excluding all charge states for a given precursor. MS2 spectra were acquired at 17500 resolutions. The data was analysed using the Proteome Discoverer 2.2 software and the spectra were searched against the NCBI protein database using the SEQUEST search engine [94]. The precursor and fragment mass tolerances were set at 10 ppm and 0.02 Da, respectively. The protease used to generate peptides, i.e. enzyme specificity was set for trypsin/P (cleavage at the C terminus of "K/R: unless followed by "P") along with a maximum missed cleavages value of two. Carbamidomethylation on cysteine was considered as fixed modification while oxidation of methionine and phosphorylation on serine, threonine and tyrosine were considered as variable modifications for database search. Peptide spectrum match and protein false discovery rate were set to 0.01 FDR.

## Determination of the efficacy of inhibitor(s) in curbing ex vivo SARS-CoV-2 infection

Vero E6 cells, considered ideal for SARS-CoV-2 infection [95], were maintained in DMEM supplemented with 10% FBS. The cells were infected with SARS-CoV-2 (Isolate USA-WA1/2020) in a biosafety containment level 3 facility. Post infection, cells were washed with phosphate buffer saline and maintained in DMEM supplemented with 2% FBS. Post 1 h infection at 400 TCID50 in 24 well plate, cells were washed with serum free media and added 500 μl of DMEM supplemented with 2% FBS containing the varying concentrations and combinations of Sorafenib, Sunitinib, SU6656 and Remdesivir in respective wells. 48 h post infection, the supernatants were collected and a quantitative RT-PCR was performed using primers of nucleocapsid gene of SARS-CoV-2 based on CDC guidelines [96]. Uninfected cells and untreated infected cells were used as negative and positive controls, respectively. The Ct values were utilized as a marker for viral replication. Statistical significance was determined using student's t-test.

## Supporting information

**S1 Fig. A multiple protein sequence alignment of NiRAN and interface domains from the RdRp molecules of coronaviruses affect various vertebrate hosts.**
(TIF)

**S2 Fig. Organization and conservation of the NiRAN Domain. (A)** Organization of the entry pockets at CoV-2-RdRp NiRAN domain lined with the strictly conserved residues**. (B)** While the conserved residues in the interface domain are scattered across the structural elements, the majority of the conserved residues in NiRAN domain lie between the antiparallel β-sheet and the immediately following helix bundle, possibly hinting at the NiRAN domain active site. **(C)** The predicted active site of the NiRAN domain possess both charged and uncharged residues, where in the charged residues primarily line the entry points of the pocket and the uncharged residues line in the deeper sections (Blue indicates positively charged regions, red indicates negatively charged regions and green indicates neutral regions, grey indicates regions beyond GTP-binding pocket).
(TIF)

**S3 Fig. Pairwise alignments corresponding to the alignment of the structural elements of CoV-2-RdRp NiRAN with known Kinases. (A)** Lim 2 kinase domain. **(B)** Syk kinase domain **(C)** O-mannosyl kinase domain. **(D)** IRAK4 kinase domain. **(E)** FGFR2 kinase domain. **(F)** Insulin receptor kinase domain. (Colour codes- Block 1: orange-cyan; Block 2: yellow-turquoise; Block 3: lime- deep blue; Block 4: green- deep blue; Block 5: green- purple).
(TIF)

**S4 Fig. The structural superimposition of the Cα chains of CoV-2-RdRp NiRAN with known Kinases reveal significant alignment of the polypeptide chains. (A)** Lim 2 kinase domain. **(B)** Syk kinase domain **(C)** O-mannosyl kinase domain. **(D)** IRAK4 kinase domain. **(E)** FGFR2 kinase domain. **(F)** Insulin receptor kinase domain. (The aforementioned kinases' Cα chains are shown in cyan, CoV-2-RdRp NiRAN Cα chains are shown in yellow.)
(TIF)

**S5 Fig. The prediction of kinase like motifs reveals the presence of sequence motifs belonging to kinase families PkA, PkC and Src. (A)** Kinase consensus site search prediction also predicts multiple phosphorylation sites. (Bold- NiRAN domain; Italics- Interface domain; Underlines: Red- Kinase consensus sequence, Blue- PkC like motif, purple- PkA like motif,

Green- Src kinase like motif, Grey- Unspecified Kinase like motifs, Yellow- Myristoylation site consensus sequence). **(B)** An inactive analog of Daidzein docked into the NiRAN domain putative active site. **(C)** An inactive analog of Geinstein docked into the NiRAN domain putative active site.
(TIF)

**S6 Fig. Purification of SARS-CoV2-RdRp. A)** Size exclusion chromatogram of recombinant SARS-CoV-2 RdRp (The peak indicated with a blue arrow represents the purified protein sample). **(B)** SDS-PAGE profile of the purified SARS-CoV-2 RdRp (The band indicated with a blue arrow represents the purified protein sample and the molecular weight markers are indicated in red arrows). **(C)** Fluorescence spectra for native and mutant SARS-CoV-2 RdRp suggest that the incorporated mutations have negligible effect on the protein structural integrity.
(TIF)

**S7 Fig. MS-MS Spectra for the peptide KSLVSKGTLVQTK from Histone H1. (A)** Histone H1 treated with SARS-CoV-2 RdRp in presence of ATP and $Mg^{2+}$. **(B)** Histone H1 treated with human Akt2 in presence of ATP and $Mg^{2+}$. **(C)** Histone H1 treated with SARS-CoV-2 RdRp in the presence of $Mg^{2+}$ only (The b and y ions are shown in red and blue, respectively).
(TIF)

**S8 Fig. Ion series detected in MS-MS for the peptide KSLVSKGTLVQTK from Histone H1. (A)** Histone H1 treated with SARS-CoV-2 RdRp in presence of ATP and $Mg^{2+}$. **(B)** Histone H1 treated with human Akt2 in presence of ATP and $Mg^{2+}$. **(C)** Histone H1 treated with SARS-CoV-2 RdRp in the presence of $Mg^{2+}$ only.
(TIF)

**S1 Table. Top 15 molecules with structurally similar fold(s) predicted by HHPred analysis.**
(XLSX)

**S2 Table. Top 15 molecules with structurally similar fold(s) predicted by ORION_DSIMB analysis.**
(XLSX)

**S3 Table. Docking analysis of kinase inhibitors at the proposed active site of NiRAN domain.**
(XLSX)

**S4 Table. The complete list of compounds selected for docking at the predicted active site of CoV-2-RdRp NiRAN domain.**
(XLSX)

**S5 Table. Docking analysis of best 5 proposed inhibitors at the proposed active site of NiRAN domain.**
(XLSX)

**S6 Table. The predicted ADME/T properties of best 5 proposed inhibitors.**
(XLSX)

**S7 Table. LC-MS/MS verification of the SARS-CoV-2 RdRp.**
(XLSX)

**S8 Table. Phosphorylated peptide fragment of protein Histone H1 (K88-K100) detected by mass spectrometry.**
(XLSX)

**S1 Data. Raw data points for the graphs presented in Figs 5A, 6A, 6B and 6C; and the raw reads for the real-time PCR analysis.**
(XLSX)

## Acknowledgments

JJ gratefully acknowledges MHRD-RUSA 2.0 [F.24/51/2014-U, Policy (TNMulti-Gen), Dept. of Edn., Govt. of India] for the infrastructure facilities provided to the Department of Bioinformatics, Alagappa University. The following reagent was deposited by the Centers for Disease Control and Prevention and obtained through BEI Resources, NIAID, NIH: SARS-Related Coronavirus 2, Isolate USA-WA1/2020, NR-52281. We thank Dr. Subeer S. Majumdar for his encouragement and support.

## Author Contributions

**Conceptualization:** Abhisek Dwivedy, Bichitra Kumar Biswal.

**Data curation:** Abhisek Dwivedy, Richard Mariadasse, Mohammed Ahmad, Sayan Chakraborty, Deepsikha Kar.

**Formal analysis:** Abhisek Dwivedy, Richard Mariadasse, Mohammed Ahmad, Sayan Chakraborty, Deepsikha Kar, Sankar Bhattacharyya, Shailendra Mani, Prafullakumar Tailor, Tanmay Majumdar, Jeyaraman Jeyakanthan, Bichitra Kumar Biswal.

**Funding acquisition:** Jeyaraman Jeyakanthan, Bichitra Kumar Biswal.

**Investigation:** Abhisek Dwivedy, Richard Mariadasse, Mohammed Ahmad, Sayan Chakraborty, Deepsikha Kar, Satish Tiwari, Sankar Bhattacharyya, Sudipta Sonar, Shailendra Mani.

**Methodology:** Abhisek Dwivedy, Richard Mariadasse, Mohammed Ahmad, Sayan Chakraborty, Deepsikha Kar, Sankar Bhattacharyya, Shailendra Mani.

**Project administration:** Prafullakumar Tailor, Tanmay Majumdar, Jeyaraman Jeyakanthan, Bichitra Kumar Biswal.

**Resources:** Sankar Bhattacharyya, Shailendra Mani, Tanmay Majumdar, Jeyaraman Jeyakanthan, Bichitra Kumar Biswal.

**Supervision:** Sankar Bhattacharyya, Shailendra Mani, Prafullakumar Tailor, Tanmay Majumdar, Jeyaraman Jeyakanthan, Bichitra Kumar Biswal.

**Validation:** Abhisek Dwivedy, Richard Mariadasse, Mohammed Ahmad, Sayan Chakraborty, Deepsikha Kar, Shailendra Mani, Prafullakumar Tailor, Tanmay Majumdar, Jeyaraman Jeyakanthan, Bichitra Kumar Biswal.

**Visualization:** Abhisek Dwivedy, Richard Mariadasse, Mohammed Ahmad, Sayan Chakraborty, Deepsikha Kar.

**Writing – original draft:** Abhisek Dwivedy.

**Writing – review & editing:** Richard Mariadasse, Mohammed Ahmad, Sayan Chakraborty, Deepsikha Kar, Satish Tiwari, Shailendra Mani, Prafullakumar Tailor, Tanmay Majumdar, Jeyaraman Jeyakanthan, Bichitra Kumar Biswal.

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
