## [Decision Letter · Decision Letter 0]

1 Apr 2021

Dear Dr. Biswal,

Thank you very much for submitting your manuscript "Characterization of the NiRAN domain from RNA-dependent RNA polymerase provides insights into a potential therapeutic target against SARS-CoV-2" for consideration at PLOS Computational Biology.

As with all papers reviewed by the journal, your manuscript was reviewed by members of the editorial board and by several independent reviewers. In light of the reviews (below this email), we would like to invite the resubmission of a significantly-revised version that takes into account the reviewers' comments.

We cannot make any decision about publication until we have seen the revised manuscript and your response to the reviewers' comments. Your revised manuscript is also likely to be sent to reviewers for further evaluation.

Sincerely,

Avner Schlessinger

Associate Editor

PLOS Computational Biology

Nir Ben-Tal

Deputy Editor

PLOS Computational Biology

Reviewer's Responses to Questions

**Comments to the Authors:**

Reviewer #1: The manuscript by Dwivedy et al. characterizes the RNA-dependent RNA polymerase from SARS-CoV-2 computationally and proposes an intriguing hypothesis that the NiRAN domain has “kinase like activity.” I cannot comment on the computational work, but as for the experimental portion there is one more important experiment lacking here, otherwise the authors cannot substantiate their claim about the activity. They measure ADP production from the RdRp using an ADP-Glo assay and conclude that there is kinase like activity. However, there are several problems with this conclusion. First, they merely measured ADP production, which implies ATP hydrolysis (or autophosphorylation), but does not show kinase-like activity. Secondly, they use the whole RdRp protein instead of the NiRAN domain. It seems possible that another part of the protein, say the RdRp active site, has ATPase activity. They need to rule out this possibility, with one or more of the following experiments:

-purify the NiRAN domain separately and show ATP hydrolysis

-show phosphotransfer activity for the whole protein or better the NiRAN domain. While no known substrates are known, standard generic substrates like dephosphorylated MBP should work.

-mutate residues in the NiRAN domain and show that the protein is still well folded but has a loss of ATP hydrolysis. They predict the active site of the NiRAN domain, but then need to test it out.

If not, then the authors can’t claim kinase like activity or any function to the NiRAN domain. While the potency of the kinase inhibitors is suggestive of kinase-like binding pocket, more needs to be shown to prove that.

Reviewer #2: This manuscript seeks to understand the function of the NiRAN domain of the SARS-CoV-2 polymerase, which is unique to nideoviruses. Based on both sequence and structural homology, the authors find that it is likely a kinase. In support of this hypothesis, they show it functions similarly to a known kinase, and that broad spectrum kinase inhibitors inhibit the activity of the NiRAN domain. Using docking, they also find some novel potential inhibitors. Together, their work provides valuable insight into this unique feature of the virus and starting points for drug discovery efforts.

Is there any evidence that the NiRAN domain is essential for the viral replication cycle? If not, it may not be a very useful drug target.

Is the active site of the NiRAN domain sufficiently different from that of other kinases to potentially allow for specific inhibition of the virus without off-target effects?

It would be useful to report docking scores and predicted binding affinities for a few suspected non-binders as a reference for judging the results for GTP, UTP, and potential inhibitors.

Many experimental screens of FDA-approved drugs have been performed. Can the authors comment on why none of these kinase inhibitors showed up as hits? Is there still hope that the NiRAN domain could be targeted successfully?

The figures are quite low resolution.

In Fig. 1, why not use the same color scheme in the structure in panel A and the secondary structure diagrams?

Reviewer #3: The manuscript by Dwivedy et al., characterizes the NiRAN domain of the RdRp protein from SARS-CoV-2 in terms of its structure, function, potential binding site and interactions with protease inhibitors. It is concluded that this domain possesses a kinase-like activity which could be inhibited by small ligands. It is therefore suggested that this domain may be a viable target for the development of new pharmaceuticals to fight of the COVID-19 pandemic.

Given the current state of the COVID-19 pandemic and the lack of therapies, any addition to the limited arsenal of tools to combat it, is important. From this perspective, the results presented in this manuscript are timely and relevant. Moreover, the manuscript is overall well written. However, there are multiple issues in the manuscript that need to be addressed before it is ready for publications. These are listed below:

Major points:

1. Line 139: Nothing that was discussed until this point actually supports the idea that the NiRAN domain assumes a kinase-like fold since its fold does not comply with the canonical kinase fold, nor with the other folds in Figure 1C.

2. Line 205: The fact that the three kinase inhibitors docked into the NiRAN site is rather meaningless as are the docking scores. You would probably be able to dock into this site other compounds which are clearly not kinase inhibitors. A more convincing argument can be put forth by searching for inactive analogues of the kinase inhibitors and demonstrating that they don’t bind well into the site.

3. Line 214: Based on the above, I don’t think that the docking analysis supports the kinase / phosphotransferase activity of NiRAN.

4. Line 251: I fail to see the value of the DFT calculations. The HOMO and LUMO energies by themselves are meaningless as are the corresponding energy differences. Similarly, the electrostatic potentials don’t add anything to the discussion. They may have been interesting had the authors tried to use them to explain the resulting binding modes, for example, by looking at how they match the electrostatic potential of the site, but this was not attempted.

5. Line 263: Why does the fact that the HOMO is more spread than the LUMO suggestive of a possible nucleophilic reaction? I would think that this spreading would reduce the electronic density at each point thereby reducing the strength of a nucleophilic attack. And which nucleophilic reaction are we talking about in the first place?

6. Line 280: Chemical reactivity towards what?

7. Line 350: Which simulations are we talking about? If this refers to the MM-GBSA simulations then their results were not discussed in the manuscript and furthermore, I don’t see what they add.

8. Line 352: Technically, 1000 nM count as nano-molar but these are really micro-molar concentrations.

9. Materials and methods: The docking protocol was not validated in any way. Table S3 contains a list of compounds selected for docking. Some of these compounds were solved in complex with their corresponding targets. The authors should try and reproduce the binding modes of these compounds with their docking protocol. Without this, how can we tell that the docking protocol is suitable for the task in hand?

10. Line 405: As stated above, what is the purpose of the MM-GBSA calculations?

11. Line 424: As noted above, I don’t see the value of the DFT calculations. Moreover, an implicit solvent model can hardly be qualified as “near physiological state” (line 431). Also, what were the starting geometries for the DFT calculations?

12. Lines 525 and 535: Docking scores by themselves are rather meaningless and are only good for comparison purposes. Furthermore, the docking scores of GTP and UTP were found to be -9.84 and -6.59 kcal/mol, respectively. Those of the three broad spectrum inhibitors were found to range between -2.6 and -3.5 kcal/mol. How could the latter then be classified as having “significantly low free binding energies”?

Minor points:

1. Line 72 and on: PDB codes should be given when discussing the cryo-EM structures.

2. Line 107: The sentence starting with “Apart from” is not clear. Are there 78 or 84 conserved residues?

3. Line 114: What is the meaning of “entry” pockets and how does the number “five” relate to the single site discussed later in the work?

4. Line 132: Change “comprises of an” to “comprises an”.

5. Line 184: This seems like a repeat of what was already discussed on line 175.

6. Line 213: I think it should be Table S4 rather than Table S3.

7. Line 339: Change “fewer” to “less”.

8. Line 355: Change “Sunitinib as” to “Sunitinib were”.

9. Line 413: The “.” Should be moved to after the “)”.

**Have all data underlying the figures and results presented in the manuscript been provided?**

Reviewer #1: Yes

Reviewer #2: Yes

PLOS authors have the option to publish the peer review history of their article (what does this mean?). If published, this will include your full peer review and any attached files.

Reviewer #1: No

Reviewer #2: No

Reviewer #3: No

**Have the authors made all data and (if applicable) computational code underlying the findings in their manuscript fully available?**

Reviewer #3: Yes
---

## [Decision Letter · Decision Letter 1]

26 Aug 2021

Dear Dr. Biswal,

Thank you very much for submitting your manuscript "Characterization of the NiRAN domain from RNA-dependent RNA polymerase provides insights into a potential therapeutic target against SARS-CoV-2" for consideration at PLOS Computational Biology. As with all papers reviewed by the journal, your manuscript was reviewed by members of the editorial board and by several independent reviewers. The reviewers appreciated the attention to an important topic. Based on the reviews, we are likely to accept this manuscript for publication, providing that you modify the manuscript according to the review recommendations.

Sincerely,

Avner Schlessinger

Associate Editor

PLOS Computational Biology

Nir Ben-Tal

Deputy Editor

PLOS Computational Biology

[LINK]

Reviewer's Responses to Questions

**Comments to the Authors:**

Reviewer #2: I'm satisfied with the author's response.

Reviewer #3: The authors have adequately responded to all my comments on the original submission and I now find the manuscript ready for publication. Still, I have found a few typos that should be corrected. These are listed below:

1. Line 47: “play” -> “playing”

2. Line 85: “hinting as” -> “hinting at”

3. Line 127: “4” -> “four”

4. Line 158: “retrieved in” -> “retrieved by”

5. Line 217: “6” -> “six”

6. Line 252: “docking in” -> “docking”

7. Line 299: “like of” -> “type”

8. Line 303: “in appropriate” -> “under appropriate”

9. Line 362: “spread across” -> “spreads across”

10. Line 399: “such Rift” -> “such as Rift”

11. Line 616: “present” -> “presents”

12. Line 648: “interaction” -> “interactions”

**Have the authors made all data and (if applicable) computational code underlying the findings in their manuscript fully available?**

Reviewer #2: Yes

Reviewer #3: Yes

PLOS authors have the option to publish the peer review history of their article (what does this mean?). If published, this will include your full peer review and any attached files.

Reviewer #2: **Yes: **Gregory R Bowman

Reviewer #3: No

Figure Files:

Data Requirements:

Reproducibility:

References:

---

## [Editor Report · Decision Letter 2]

26 Aug 2021

Dear Dr. Biswal,

We are pleased to inform you that your manuscript 'Characterization of the NiRAN domain from RNA-dependent RNA polymerase provides insights into a potential therapeutic target against SARS-CoV-2' has been provisionally accepted for publication in PLOS Computational Biology.

Best regards,

Avner Schlessinger

Associate Editor

PLOS Computational Biology

Nir Ben-Tal

Deputy Editor

PLOS Computational Biology

---

## [Editor Report · Acceptance letter]

31 Aug 2021

PCOMPBIOL-D-20-02312R2 

Characterization of the NiRAN domain from RNA-dependent RNA polymerase provides insights into a potential therapeutic target against SARS-CoV-2

Dear Dr Biswal,

I am pleased to inform you that your manuscript has been formally accepted for publication in PLOS Computational Biology. Your manuscript is now with our production department and you will be notified of the publication date in due course.

With kind regards,

Zsofi Zombor
